# The Therapeutic Mechanisms of Mesenchymal Stem Cells in MS—A Review Focusing on Neuroprotective Properties

**DOI:** 10.3390/ijms25031365

**Published:** 2024-01-23

**Authors:** Sonia Gavasso, Torbjørn Kråkenes, Håkon Olsen, Elisabeth Claire Evjenth, Marie Ytterdal, Jonas Bull Haugsøen, Christopher Elnan Kvistad

**Affiliations:** 1Department of Clinical Medicine, University of Bergen, 5009 Bergen, Norway; torbjorn.krakenes@helse-bergen.no (T.K.); hakon.olsen@helse-bergen.no (H.O.); elisabeth.evjenth@student.uib.no (E.C.E.); jonas.haugsoen@uib.no (J.B.H.); christopher.kvistad@helse-bergen.no (C.E.K.); 2Neuro-SysMed, Department of Neurology, Haukeland University Hospital, 5021 Bergen, Norway

**Keywords:** multiple sclerosis, mesenchymal stem cells, stem cell therapy, neuroprotection, immunomodulation, cell migration, paracrine signaling, microglia, inflammation, autoimmunity, ferroptosis, remyelination, neuroregeneration

## Abstract

In multiple sclerosis (MS), there is a great need for treatment with the ability to suppress compartmentalized inflammation within the central nervous system (CNS) and to promote remyelination and regeneration. Mesenchymal stem cells (MSCs) represent a promising therapeutic option, as they have been shown to migrate to the site of CNS injury and exert neuroprotective properties, including immunomodulation, neurotrophic factor secretion, and endogenous neural stem cell stimulation. This review summarizes the current understanding of the underlying neuroprotective mechanisms and discusses the translation of MSC transplantation and their derivatives from pre-clinical demyelinating models to clinical trials with MS patients.

## 1. Introduction

### 1.1. Multiple Sclerosis

Multiple sclerosis (MS) is an immune-mediated disease of the central nervous system (CNS) characterized by inflammation, causing multifocal demyelination and subsequent neuronal degeneration. Globally, some 2.8 million people are affected, and the prevalence has increased in recent decades, making MS the most common non-traumatic cause of disability in young adults [1].

Traditionally, MS has been considered a disease triggered by T cell-mediated autoimmune events with a breakdown of the blood–brain barrier (BBB) and peripheral immune cells invading the brain parenchyma, causing inflammation, demyelination, and secondary neuronal loss. Recent data show that infection with the Epstein–Barr virus (EBV) plays a crucial role in initiating pathogenic immunological events in MS [2]. In a large cohort comprising 10 million young adults, longitudinal data revealed that the risk of MS increased 32-fold after infection with EBV, but not after infection with other viruses. Molecular mimicry has also been identified between the EBV transcription factor EBV nuclear antigen 1 (EBNA1) and the CNS protein glial cell adhesion molecule (GlialCAM) [3].

The beneficial effects of anti-CD20 therapies point to a central role of B cells in the pathogenic cascade. In the later stages of the disease, neurodegeneration leads to loss of brain tissue and atrophy. Many lesions may also continue to expand slowly. These chronic active lesions, also called smoldering lesions, contain an expanding ring of microglia surrounding the inactive demyelinated area and may be responsible for the continuous progression of disability [4].

All parts of the CNS may be affected by MS, and common symptoms are motor and sensory deficits, visual impairment, and cognitive dysfunction. The different clinical subtypes of MS are historically characterized by substantial differences in response to immunomodulating therapies. While the inflammatory component of relapsing-remitting MS (RRMS) can be controlled by highly effective immune modulation, chronic compartmentalized inflammation and persistent neurodegeneration dominate the primary and secondary progressive forms of MS (PPMS and SPMS). The prevention of this degeneration and promotion of remyelination and axonal regeneration represent major hurdles in today’s MS therapy.

### 1.2. Mesenchymal Stem Cells

Mesenchymal stem cells (MSCs) are heterogeneous cells with self-renewal potential and multipotent properties. MSCs do not have a unique cell marker, but are defined according to international guidelines by the presence and absence of different cell surface proteins and tri-lineage differentiation potential in vitro [5].

In contrast to neural stem cells, MSCs can be obtained from different tissues, such as bone marrow (BM), adipose tissue, or umbilical cord, and expanded ex vivo. The use of autologous or allogeneic MSCs represents no ethical concerns in contrast to other stem cell therapies based on embryonal or fetal stem cells. Genetic manipulation is also unnecessary, as is the case with induced pluripotent stem cells. This, along with their intrinsic role in tissue repair, has made MSCs an attractive candidate for human trials. Several murine studies underline their potential regenerative role in MS, as MSCs have led to increased remyelination and improved outcomes in different disease models [6].

Several open-label trials have reported promising results supporting the safety and feasibility of MSC treatment in MS patients, and some have also shown beneficial clinical effects [7,8,9,10,11]. As MSCs may represent a promising therapeutic modality for MS, there is a need for a greater understanding of the underlying mechanisms of action. This narrative review will focus on these mechanisms, as outlined in Figure 1, and summarize the current knowledge at hand. We will also discuss the status for the translation of pre-clinical results into clinical trials.

## 2. Mechanisms

### 2.1. Paracrine Function

MSCs are highly secretory, and a substantial part of their regenerative potential has been attributed to paracrine functions. Several neurotrophic growth factors are secreted from MSCs, such as nerve growth factor (NGF), brain-derived neurotrophic factor (BDNF), hepatocyte growth factor (HGF), and vascular endothelial growth factor (VEGF) [12]. These proteins increase neuronal proliferation and survival, as well as endogenous neurogenesis. The expression levels of both BDNF and NGF have been shown to correlate with the ability of MSCs to promote neuronal survival and neurite outgrowth [13]. In an in vitro study of rat cortical and hippocampal neurons, the axonal outgrowth diminished when BDNF was depleted from the secretome [14]. Furthermore, the anti-apoptotic effect of BM-MSCs on motor neurons exposed to acetylacetone was eliminated when NGF was removed from the secreted factors [15]. These findings point to NGF and BDNF being essential parts of the MSCs neuroprotective capacity, although other studies have shown different results, which will be discussed in a later section [16,17].

Several in vivo studies applying different MS models have demonstrated the potent paracrine effects of MSCs. In a study using the experimental autoimmune encephalomyelitis (EAE) model, mice receiving BM-MSCs showed improved clinical and histological outcomes when the MSCs were administrated in the early or middle phase of the disease [18]. The MSCs ameliorated inflammation by suppressing T cell activation and by inducing T cell anergy. The conditioned medium from the MCSs had similar effects as the cells themselves, highlighting the significance of the paracrine mechanism of action.

The paracrine functions of the MSCs are mediated through secreted molecules, collectively named the secretome, which have been shown to contain several diffusible biomolecules that promote the development, maintenance, repair, and survival of neuronal populations [19]. The soluble fraction contains cytokines and chemokines, such as IL-10, IL6 and CXCL-10 and growth factors, including GDNF, FGF, IGF and BDNF [20]. Beyond the soluble molecules, the secretome also contains extracellular vesicles with cargo such as proteins, nucleic acids, lipids and metabolites. The components of the secretome, and their effect on the environment, may vary depending on the source of MSCs and if pre-conditioning strategies have been applied.

In an EAE mouse model, applying the secretome of stem cells from human exfoliated deciduous teeth (SHED) led to improved disability scores and a reduction in inflammation, demyelination, and axonal injury [21]. The SHED secretome effectively inhibited T cell proliferation and reduced the production of pro-inflammatory cytokines. In addition, the infiltrating macrophages shifted from a pro-inflammatory phenotype to a pro-regenerative phenotype, thereby improving outcomes. A recent in vitro study also showed that the secretome from MSC-derived neural progenitor cells reduced the expression of pro-inflammatory markers in activated microglia [22]. Other in vitro studies have revealed that the paracrine function of MSCs influences the destiny of neural stem cells by enhancing oligodendrogenesis and neurogenesis [23,24]. This may lead to increased remyelination in MS.

A sub-group of the extracellular vesicles in the secreome is called exosomes. These membrane-coated particles are 30–150 nm in diameter and transport different proteins and nucleic acids, serving as paracrine mediators. More than 300 proteins and 150 microRNAs have been identified in the exosomes of MSCs, in addition to other biomolecules [25]. The exosomes may be isolated by using ultracentrifugation in combination with differential centrifugation or cross-flow filtration [26]. Exosomes may fully recapitulate, and even improve, the therapeutic effects of MSCs with regard to immunomodulation, stimulation of neurogenesis, and inhibition of apoptosis [27]. Exosomes have also been shown to have a modulatory effect on activated microglia in MS animal models. In a study using the EAE model, mice receiving intravenously administered exosomes had decreased activation of microglia and reduced inflammation and demyelination, resulting in improved functional outcomes [28]. In a similar study applying the EAE disease model, exosomes from MSCs were administrated intranasally. Results showed that nasal exosome treatment decreased CNS inflammation more effectively than treatment with MSCs, leading to better amelioration of the disease [29].

### 2.2. Remyelination

Although there is some spontaneous remyelination in MS patients, this mainly occurs in the early stages of the disease, and the newly generated myelin is inferior to the normal myelin, being thinner and more fragile [30]. In the chronic phases of the disease, the remyelination capacity is considerably reduced due to poor recruitment of oligodendrocyte precursor cells (OPCs) and failure of the OPCs to differentiate into mature oligodendrocytes [31]. MSCs have shown a potential to promote remyelination in different MS models. The secreted factors of MSCs can activate oligodendrogenesis in postmitotic neural progenitor cells by boosting oligodendroglial differentiation and maturation at the cost of astrogenesis [16]. Similar results were shown in vivo using the EAE model, as MSCs enhanced oligodendrocyte differentiation and remyelination [24,32]. The MSC secretome also promoted the differentiation of oligodendrocytes and neurons in an EAE model, resulting in improved outcomes [33]. Hepatocyte growth factor (HGF) was identified as an important contributor as depletion of HGF blocked functional recovery, whereas HGF alone mediated similar beneficial effects. In another study applying the EAE model, transplanted MSCs decreased oligodendrocyte apoptosis, thus reducing demyelination, and improving functional recovery [34].

The stimulating effect of MSCs on OPCs has also been tested in a non-inflammatory model where MSCs were co-transplanted with OPCs into a myelin-deficient mouse strain [35]. MSCs increased migration, engraftment, and maturation of myelinating oligodendrocytes, producing robust myelination in the corpus callosum. Several neurotrophic growth factors, including HGF, could not reproduce the remyelinating effect of the MSCs alone [16,17,36]. This suggests that other molecules and mechanisms, such as microRNAs, mediate the pro-regenerative effects of MSCs. These short, non-coding RNA molecules regulate gene expression by binding to target mRNA. MicroRNAs have been shown to play an important role in the regulation of remyelination by affecting the phagocytic activity of microglia and promoting oligodendrocyte maturation [37]. The secretome of MSCs contains a plethora of microRNAs packed in extracellular vesicles, including those affecting demyelination and remyelination. In an EAE model, BM-MSC exosomes carrying miR-367-3p prevented inflammation and demyelination to a greater extent than controls via inhibition of microglia death, suggesting a potential neuroprotective effect [38]. Likewise, an in vitro model of ischemic stroke showed a positive effect of miR-134 obtained from BM-MSCs on oligodendrocytes by suppressing apoptosis. This ability would also be highly relevant from an MS perspective [39].

MSCs have also shown an ability to promote remyelination in toxic demyelinating models. In a recent study, MSCs increased oligodendrocyte numbers and myelin levels in a cuprizone mouse model, presumably by reducing mitochondrial dysfunction [40]. Furthermore, another study applying the same cuprizone model showed that both human BM-MSCs and SHEDs injected intraperitoneally decreased demyelination and microglial inflammation [41]. Other studies have, however, shown negative results regarding remyelination in both EAE [42] and cuprizone [43] models after treatment with MSCs. Potential reasons for these discrepancies include differences in mode, type, dosage, and timing of MSC administration. The results may also highlight the heterogeneous nature of these stem cells.

### 2.3. Immunomodulation of the Adaptive Immune System in MS

Acute inflammation in the CNS is detrimental due to immune activation, resulting in edema and cell death, as observed in focal lesions in MS. Similarly, chronic inflammation in the CNS can lead to persistent neurodegeneration and hinder regenerative processes, as seen in chronic active lesions in MS. The CNS is known for its immune tolerance, as demonstrated by transplantation outcomes, but it can also exhibit immune responses, as observed in its reaction to viral infections [44]. Additionally, immune surveillance occurs at the CNS borders in the meninges [45], and there is growing support for the idea that the skull bone marrow may play a role in brain inflammation and pathologies [46]. The immune response in the CNS lacks the regenerative capability of the peripheral nervous system, where Schwann cells and tissue signals such as TGF-β play a critical role [47,48]. This immune privilege of the CNS tissue leads to its vulnerability to prolonged inflammation and immune activation, similar to the eyes and testes, where inflammatory processes lead to tissue damage and functional deficits with scarring rather than repair. Notably, and potentially relevant for MS, therapeutic effects of stem cells may be preventative, as demonstrated in an animal model of preterm brain injury where early interventions protected brain structures, reduced the inflammatory response by increasing IL-10 and reduced oxidative stress and lipid peroxidation [49].

Inflammation is a double-edged sword controlled by tissues to maintain a balance between function, immunity, and repair in response to damage [50,51]. Inflammation is necessary to clear pathogens and cellular debris and, most importantly, for tissue repair and regeneration. Tissue-specific growth factors, cytokines, and chemokines attract repair-promoting cells such as MSCs and modulate their function based on tissue-specific demands, often through receptors like Toll-like receptors (TLRs) [50,51]. The immune system and the regenerative system are activated in response to alarm signals from injured tissue, and these pattern recognition receptors, such as TLRs, are crucial for immune activation and mobilization of the repair response [52,53].

In the context of MS, the fundamental approach of disease-modifying treatments (DMTs) revolves around depleting and inhibiting components of the peripheral adaptive immune system, aiming to halt the formation of new focal lesions within the CNS. The primary objective of these treatments is to mitigate further inflammation, thereby impeding the progression of the disease. However, it is important to note that these interventions do not inherently enhance the regenerative capabilities of the compromised CNS tissue. MSCs offer a potential avenue for immunomodulation in MS, with the aim of fostering regenerative processes.

In EAE, MSCs have demonstrated the ability to control disease progression through various mechanisms. These include the induction of regulatory T cells while suppressing pathogenic Th1/Th17 phenotypes, upregulation of suppressive and regulatory cytokines such as TGF-β and IL-10, as well as the release of exosome miRNAs such as miR-let7. Simultaneously, inflammatory molecules such as IL-17 and IFN-γ are downregulated [54]. Intriguingly, in vitro studies using human peripheral blood CD4+ T cells from Parkinson’s disease patients have indicated that MSCs can induce a shift towards regulatory T cells and suppress Th17 responses, possibly mediated through direct cell-to-cell contact [55].

In a clinical study involving patients with RRMS, the use of MSCs showed promising results. The study observed a reduction in inflammation on MRI and a tendency towards decreased levels in pathogenic inflammatory Th1 and Th17 cell subtypes. Notably, an increase in regulatory B cells was also observed [56]. Further in vitro investigations revealed that MSCs exerted similar immunomodulatory effects on peripheral blood mononuclear cells from MS patients. These effects were mediated by an increase in regulatory B cells through the indoleamine 2,3-dioxygenase (IDO) pathway, which is well-known for its role in T cell arrest and regulation [57]. While direct cell-to-cell contact may be important for some of the immunosuppressive effects of MSCs, they also release potent immune modulators such as PD-L1/2 ligands, IDO, IL-10, TGF-β, miRNAs, and prostaglandins. In the context of MS, dendritic cells play a central role in immune activation, and MSCs have demonstrated the ability to modulate the activity of dendritic cells both in vitro and in vivo, thereby influencing the immune response [58,59].

In another clinical trial including 15 patients with RRMS who had not responded to conventional DMTs, MSCs exhibited systemic effects on the immune system. This was evidenced in the increased proportion of regulatory T cells and a decrease in the proportion of activated myeloid dendric cells and lymphocytes. Intriguingly, these MSC-induced effects persisted in vitro, with a decrease in lymphocyte proliferation observed in immune cells from treated patients [58]. Clinically, there was a decline in the mean expanded disability status scale (EDSS) scores and no new MRI lesions observed at six months follow-up, supporting the efficacy of MSC therapy. Remarkably, this study employed ferumoxides to label MSCs, allowing for their detection by MRI, which revealed their dissemination from lumbar site of inoculation to various CNS regions, including the occipital horns, meninges, spinal roots, and spinal cord parenchyma.

However, in the context of SPMS, MSC did not appear to modulate peripheral T cell subsets or humoral immunity to common antigens, despite showing evidence of neuroprotection [9]. This suggests that the mechanisms underlying the effects of MSCs in RRMS, and progressive MS may differ significantly. It underscores the notion that MSCs exert variable immunomodulatory effects on different types of immune cells, contingent upon the local microenvironment, disease status and phenotype [53].

The existence of varying immune mechanisms of MSCs based on disease phenotype is further substantiated by clinical evidence in other autoimmune diseases, such as systemic lupus erythematosus. In this context, MSCs exhibited positive effects in patients with an inflammatory phenotype characterized by elevated levels of IFN-γ and low levels of IL-6. However, this beneficial effect was not observed in patients with low inflammation [60]. MSCs seem to possess the ability to adapt their functional capabilities to meet the demands of different damage or disease phenotypes, possibly involving damage-associated recognition receptors like TLRs. For example, in co-cultures of human TLR3-primed MSCs with peripheral mononuclear blood cells activated by CD3/CD28 stimulation, the primed MSCs induced immune suppression. In contrast, TLR4-primed MSCs promoted inflammation [61]. In a further study, TLR3-primed MSCs induced regulatory T cells through the release of suppressive TGF-β and the presence of double-stranded RNA and high levels of TNF-α and INF-γ. In contrast, TLR4-primed MSCs induced the release of chemokines and activated T cells in the presence of lipopolysaccharide [52]. Additionally, in vitro experiments revealed that MSC upregulated and secreted Programmed Death-1 receptors (PD-1) upon IFN-γ and TNF-α stimulation. These receptors are well-known checkpoint inhibitors for regulating and suppressing T cell activation, highlighting another mechanism indicative of immune suppression by MSCs [62]. Thus, the therapeutic effects of MSCs in different MS phenotypes may indeed be distinct, emphasizing the importance of identifying the underlying mechanisms essential for optimizing MSC and MSC-derived therapies.

### 2.4. Immunomodulation of Microglia in MS

Microglia are primary innate immune cells, and function as the resident tissue macrophage in the CNS. The functions of microglia are highly varied, both in homeostasis and in disease, and include synaptic pruning, phagocytosis, immune surveillance, injury response, and secretion of neurotrophic factors. Microglia are typically divided into two groups based on their function. The classically activated microglia (M1) are induced by pathogens and pro-inflammatory factors. Functionally, M1 microglia contribute to a pro-inflammatory environment by secretion of reactive oxygen species and cytokines such as TNF-α, IL-1β, and IL-6 [63,64,65]. M2 microglia, on the other hand, promote the release of anti-inflammatory factors such as IL-10 and TGF-β [63]. They also stimulate regeneration by secretion of the growth factors IGF-1, FGF, and CSF-1, in addition to neurotrophic factors such as NGF and BDNF [63,66]. Multiple studies show that microglia phenotyping is multidimensional, with extensive overlap in gene expression, rather than the classically simple linear spectrum [67,68,69].

A shift between M1 and M2 microglia may be necessary for neuroregeneration, and MSCs represent a promising avenue in this regard. In an in vitro microglia model using lipopolysaccharide-induction of BV-2 cells, MSCs significantly inhibited the expression of pro-inflammatory mediators in M1 microglia [70]. This effect was attenuated when tumor necrosis factor-inducible gene 6 protein (TSG-6) was silenced, suggesting that MSCs modulate microglia activation through TSG-6. Moreover, MSCs also promoted M2 polarization via TSG-6 both in vitro and in vivo [71]. Thus, TSG-6 seems to play a key role in the MSC-mediated microglial modulation, which is highly relevant for neuroinflammatory diseases, such as MS.

There are also other mechanisms relevant to the effect of MSCs on microglia. In a chronic cuprizone mouse model, intraventricular injections of MSCs reduced microglia and astrocyte activation and ameliorated inflammation by secretion of the trophic factors CX3CL1 and TGF-β [72]. This suggests that MSCs may have a beneficial effect on chronically activated microglia, which is interesting from a progressive MS perspective. Another study showed that MSCs reversed microglia activation in vitro by the secretion of colony-stimulating factor 1 [73]. The microglia treated with MSCs and MSC conditioned medium also increased the production of neurotrophic and neuroprotective factors such as ADNP, BDNF, and FGF2, in addition to Arginase-1, a marker for M2 microglia. Moreover, the phagocytotic activity of the microglia was drastically increased in the MSC-treated group, which may be relevant for a regenerative perspective as the phagocytosis of myelin debris is necessary for the initiation of remyelination within the CNS.

### 2.5. Migration

The therapeutic potential of MSCs in treating MS is intricately tied to their ability to migrate toward areas of damage or inflammation within the CNS. The migration of MSCs is a complex process influenced by various factors, including cellular interactions, signaling mechanisms, and the pathological environment. This process can be understood in two contexts: systemic and local mechanisms.

In systemic homing, MSCs enter the bloodstream through deliberate administration or natural recruitment, followed by sequential interactions that guide them to the injury site. In homing to tissues, MSCs employ mechanisms like those of leukocytes. The initial step involves MSCs tethering and rolling on the endothelial cell surface, mediated by interactions between endothelial cell selectins and MSC-expressed ligands. Despite the lack of typical ligands, such as HCELL and PSGL-1 on MSCs, alternative ligands, like Galectin-1 and potentially CD24, are thought to play an important role in this interaction [74,75,76,77,78]. The subsequent activation phase is driven by G protein-coupled chemokine receptors on MSCs responding to inflammatory signals. In this context, the role of SDF-1 expressed by endothelial cells and its binding to CXCR4 receptors on MSCs is significant [79,80,81,82,83].

Following activation, MSCs engage in firm adhesion to the endothelium, which involves integrins such as VLA-4 on MSCs binding to endothelial VCAM-1 [84]. Post-activation chemokines like SDF-1 (CXCL12) are pivotal in this phase. The complexity of MSC migration is further enhanced by the potential expression of various adhesion molecules by MSCs themselves [85,86,87]. The next step, transmigration, or diapedesis, requires MSCs to penetrate the endothelial layer and the basement membrane. This is facilitated by matrix metalloproteinases and modulated by tissue inhibitors of metalloproteinases in response to the inflammatory environment [88,89].

Finally, MSCs are directed toward the injury site by chemotactic signals, including growth factors and chemokines like PDGF-AB, IGF-1, and SDF-1, with inflammatory mediators like TNF-α enhancing this migratory response [90,91]. MSCs have an inherent tropism for sites of inflammation, which is crucial in the context of MS pathology. Chemokine receptors such as CCR2, CCR3, and CXCR4 guide MSC migration toward lesions in response to chemokines like CCL2 and SDF-1 [92,93]. Additionally, MSC migration involves the activation of signaling pathways, like PI3K/AKT and MAPK, which respond to environmental cues and are essential for directed movement toward injury sites.

However, the use of MSCs in MS faces several challenges, particularly related to their systemic administration. A significant hurdle is the pulmonary first-pass effect, where most intravenously administered MSCs become entrapped in the lungs [94]. This phenomenon can limit the number of cells that reach the CNS, potentially reducing the treatment’s effectiveness. Strategies such as modifying MSC surface properties to reduce lung entrapment or utilizing localized delivery methods are being explored to mitigate this issue. Another challenge is the variability in MSCs’ expression of homing receptors, which can impact their migratory efficiency. Advanced cell sorting and preconditioning techniques are being investigated to select and enhance MSC populations with higher expression of relevant homing receptors.

### 2.6. Horizontal Information Transfer and Alleviation of Ferroptosis

Horizontal information transfer is a fundamental process in cellular communication, facilitated by structures like gap junctions and tunneling nanotubes (TNTs). These structures enable the exchange of various molecules, including proteins, nucleic acids, and organelles, such as mitochondria [95,96]. This exchange of information plays a crucial role in coordinating tissue functions and maintaining homeostasis, particularly in situations where inflammation hampers tissue function and needs resolution [97].

Recent research indicates that the therapeutic effects of administered MSCs extend beyond the mere secretion of molecules within extracellular vesicles to include direct cell-to-cell contacts, such as TNTs and gap junctions [98]. Of particular interest is MSCs’ ability to transfer mitochondria to injured or stressed cells, restoring or enhancing their functional activity to support a robust regenerative response. In an in vivo model of acute respiratory distress syndrome (ARDS) and sepsis, the transfer of mitochondria from MSCs to macrophages via TNTs enhanced their phagocytic activity [99]. In another study, MSCs repaired postischemic endothelial cells by transferring functional mitochondria via TNTs in vitro [100]. Moreover, TNT-mediated mitochondrial transfer improved recovery from ischemic stroke in a rat model [101]. Mitochondrial transfer from MSCs via TNTs could also enhance the survival of damaged neuronal stem cells [102,103]. In cell co-culture systems, MSCs and neurons establish direct connections through gap junctions, TNTs, and indirect communication via extracellular vesicles [95]. These mechanisms are highly relevant in the context of MS, where mitochondrial dysfunction has been identified as a key contributor to neurodegeneration [104]. Interestingly, this flow of information is unidirectional.

MSCs may also be able to protect neurons and glial cells by preventing ferroptosis. Ferroptosis is a regulated oxidative cell death characterized by iron-driven lipid peroxidation, resulting in oxidative stress and the production of harmful reactive oxygen species (ROS) that affect mitochondrial redox homeostasis [105,106]. Iron dysregulation and myelin integrity are implicated in the pathogenesis of MS, aging, and neurodegenerative diseases [107]. MRI studies revealed a possible negative association between iron content and myelin content [108]. In MS, iron accumulation progressively increases from RRMS to progressive MS and correlates with persistent neuro-inflammation, activated microglia, neuronal degeneration, and demyelination. In fact, a hallmark of slowly expanding lesions that are more common in progressive MS is a rim of activated iron-containing microglia, a feature associated with worse clinical disability scores [109]. The presence of elevated levels of labile iron and peroxidized phospholipids in active and chronic lesions, as well as in the CSF of MS patients, are signs of ongoing ferroptosis and iron dysregulation that can lead to auto-amplifying inflammation and cell death. Notably, ferroptosis has been shown to drive T cell immune-mediated neurodegeneration in MS [110,111]. In the cuprizone animal model of demyelination and remyelination, oligodendrocyte loss caused by ferroptosis could be inhibited by treatment with ferrostatin-1, an inhibitor of ferroptosis; interestingly, iron localization shifted from oligodendrocytes to macrophages which could potentially explain the progression of slowly expanding lesions in MS [112].

A recent EAE study demonstrated that ferroptosis in microglia could be suppressed and symptoms alleviated by treating animals with MSC-derived exosomes containing miR-367-3p [38]. The underlying mechanisms of ferroptosis suppression may be attributed to miRNA inhibiting Enhancer of zeste homolog 2 (EZH2) and activation of Glutathione Peroxidase 4 (GPX4), a major ferroptosis inhibitor. Several other recent studies have shown the potential beneficial effects of ferroptosis inhibitors in animal models of MS [110,111].

Another recently discovered neuroprotective mechanism of MSCs involves mitigating ferroptosis in neurons through mitochondrial transfer via TNTs, thus restoring mitochondrial function [113]. MSCs’ mitochondrial transfer has demonstrated the ability to rescue cisplatin-damaged neuronal stem cells in vitro and through intranasal administration in an animal model [103]. A potent but undesired effect of the stem cell rescue or protection of MSC mitochondrial transfer has been shown in glioblastoma, where MSCs are recruited to the cancer and confer chemoresistance, underscoring its context-dependent nature [102]. Iron dysregulation and ferroptosis have also been implicated in neurodegenerative diseases such as Parkinson’s and ALS. Researchers are exploring the potential of MSCs and their derivatives to alleviate these processes in these diseases as well [114].

### 2.7. Transdifferentiation

In vitro studies have shown that MSCs can transdifferentiate into neural-like cells [115,116,117]. The MCSs change morphology resembling neural stem cells concentrated in hubs with axon-like arms connecting the cells. This may be seen after modification of the culture medium with different neural growth factors, such as EGF and FGF. The neural differentiated MSCs express neuroglial markers, including nestin, SOX2, neurofilament light chain, and TAU proteins. Application of CSF to the growth medium also promotes transdifferentiation into a neural phenotype [118]. A recent study showed that MSCs from different tissues, including bone marrow, differentiated towards a neural phenotype through a dedifferentiation step [115]. With a neural induction medium, the cells gradually adopted a complex morphology with dendrite- and axon-like structures. However, no studies have so far been able to show electrophysiological activity in trans-differentiated MSCs, which is an essential characteristic of mature neurons.

Studies have indicated that MSCs may also transdifferentiate into a neural phenotype in vivo. In a rodent EAE model, human BM-MSCs were found in inflamed spinal cord regions 43 days following intravenous injections [119]. A small number (<1%) of the infiltrated MSCs expressed the neural markers nestin and beta-tubulin III, suggesting that some cells were able to transdifferentiate in a neural direction. Mice treated with BM-MSCs had reduced demyelination and improved outcomes. These findings align with another study performed in an EAE model of primates [120]. Intrathecally administered human embryonic stem cell-derived MSC spheres attenuated disease progression and prevented demyelination following a three-month observation period. The cells were located within the CNS, and some expressed the neural stem cell markers Stem121 and MBP, indicating transdifferentiation. It was, however, not possible to exclude that the MSCs could have fused with the neurons already present, thereby gaining a neural phenotype. The small amount of MSCs with neural markers also suggests that other mechanisms are responsible for their therapeutic efficacy.

### 2.8. Priming of MSCs

The therapeutic abilities of MSCs and their secretome may be modified and enhanced by priming or preconditioning in vitro prior to administration. This preconditioning includes physical, chemical, and biological modification of the culture medium and/or environment [121]. Hypoxia is a widely investigated example of such preconditioning: In the human body, MSCs are in a microenvironment with low partial pressure of oxygen, usually between 1 and 5%, whereas a cell culture environment contains around 20% [122]. It is thus important to assess how hypoxia preconditioning affects MSCs and their therapeutic properties.

Giacoppo et al. investigated if secretomes from human periodontal ligament stem cells preconditioned with hypoxia could improve progression in an EAE mouse model of MS [123]. Results from disease scoring showed that secretome-treated mice improved significantly as compared to untreated EAE mice, with decreased levels of proinflammatory cytokines and upregulation of the anti-inflammatory cytokine IL-37. A reduction in oxidative stress and apoptosis was also shown. Another study investigated how secretomes from human BM-MSCs cultured under 0.1%, 5%, or 21% oxygen partial pressure affected MSCs’ paracrine functions [124]. After culturing the MSCs for 14 days with 0.1% oxygen partial pressure, the secretome exhibited enhanced chemotactic and proangiogenic properties.

Preconditioning MSCs with modified culture media is another method used to enhance their differentiation potential. Priming human umbilical cord MSCs (hUCMSCs) with IFN-γ in a rodent EAE model improved the alleviation of neurological deficits as compared to hUCMSCs cultured in normal conditions [125]. Also, the concentrations of IL-17A and TNF-α were lower in the primed MSCs, underlining the potential of improving the anti-inflammatory capabilities of MSCs by this sort of priming.

The therapeutic potential of MSCs may also be potentiated by supplements of neural growth factors to the culture medium. In an EAE mouse model, MSC-derived neural progenitors (MSC-NPs) were differentiated from BM-MSCs through a neural medium supplementation of EGF and bFGF and administered intrathecally. This resulted in reduced immune cell infiltration, decreased demyelination, and improved neurological function [126]. This approach has also reached the stadium of clinical trials. In a phase I clinical trial, MSC-NPs were administered to 20 patients with progressive MS. The treatment was safe and showed promising results with improved median EDSS scores, suggesting possible efficacy [127]. This sort of preconditioning strategy permits the customization and “hand-tailoring” of MSCs and their secretome into specific anti-inflammatory and neuroregenerative directions, thereby increasing the therapeutic potential in MS.

## 3. Discussion

High-intensity DMTs, such as B-cell depleting therapies and hematopoietic stem cell transplantation (HSCT), have proven effective in preventing disease activity in RRMS. Currently, no treatment is available to suppress compartmentalized inflammation, which plays a significant role in disease progression. Additionally, direct promotion of regeneration remains an unmet need. As demonstrated in this review, MSCs have the potential to address both of these aspects through several different mechanisms.

Although some studies have shown findings suggesting in vivo implantation and transdifferentiation of MSCs, an increasing amount of evidence points to secretory functions and cell-to-cell communicating abilities as the most important mechanisms responsible for their therapeutic impact. Through these mechanisms, MSCs contribute indirectly to endogenous neuroprotective and regenerative processes. Specifically, MSCs have been shown to promote CNS regeneration via modulation of microglia, remyelination by enhanced differentiation of oligodendrocytes, and improved neural survival and outgrowth. The ability of the MSCs to migrate towards lesions within the CNS is also a clear advantage in MS, as direct cell injection into MS lesions is invasive and difficult to achieve considering the multifocality of the disease.

Although MSCs have shown beneficial results in in vitro and in vivo models of demyelination and axonal injury, this does not necessarily mean the same applies to the human CNS. Several clinical trials have been performed testing MSC transplantation in MS (Table 1), and trials are ongoing (Table 2). Although most studies have shown safety and clinical improvement in a subset of patients, the studies differ in inclusion criteria, endpoints, MSC administration, type of MSCs, method of culturing, and study design, which make the results hard to combine or compare. Trials showing the most promising results have a small number of participants and lack a control group, complicating the interpretation of the findings.

Recently, a large clinical trial including 144 patients failed to show decreased disease activity in patients receiving autologous BM-MSCs intravenously as compared to placebo [128]. This appears in contrast to the high number of rodent MS models, where systemically administrated MSCs consistently have reduced inflammation and promoted remyelination and axonal regeneration. Several points may explain this discrepancy. First, as MS is a disease only occurring in humans, there is no completely satisfactory in vivo MS model. Neither the EAE nor the cuprizone model comprise the complete pathophysiology of MS. This may cause problems translating successful pre-clinical results into MS patients. From a neuroregenerative point of view, however, one could argue that the mechanisms responsible for axonal repair and remyelination are relatively universal, regardless of pathophysiology and disease model.

Secondly, MSCs represent a heterogeneous population that lacks a specific cell marker. There are, in other words, bound to be minor variations from batch to batch and person to person, which may affect therapeutic efficacy. BM-derived MSCs from MS patients have also shown a reduced proliferative capacity and accelerated cellular aging compared to BM-MSCs from healthy persons [120]. This may also influence the therapeutic potential of autologous transplantation in MS patients. Similarly, cryopreservation can also impact the MSCs, which can lead to altered clinical effects [129]. Most clinical trials have used cryopreserved MSCs.

Third, in many rodent studies showing beneficial results, the number of administered MSCs has proportionally been far higher than in clinical trials. Typically, one million MSCs per mouse have been injected intravenously or intraperitoneally [41]. As mice weigh 20 g, the equivalent dose for a 70 kg human would be 3.5 billion MSCs, which is inconceivable. With this backdrop, it may not be surprising that a trial with an intravenous dose of 1–2 million MSCs/kg bodyweight failed to show the same positive effects [128].

The mode of administration is also important. Although rodent studies have shown that MSCs can migrate into the CNS following systemic administration, albeit in small numbers, most MSCs become trapped in the lungs shortly after injection [94]. From this location, it is unlikely that the MSCs can provide neuroprotective effects in MS. Trials assessing intrathecally delivered MSCs have generally shown more promising results than those applying the intravenous route [130]. Recently, the results of the first placebo-controlled trial using intrathecal administration were published [131]. This cross-over designed study included patients with active progressive MS and showed that more patients treated with MSCs exhibited no evidence of disease activity and improved disability scores as compared with the sham-treated group. Patients receiving MSCs also had improved functional tests, including T25FW, 9-HPT, and changes in motor networks on functional MRI, which indicate neuroprotection. Neurofilament light chain (NF-L), a biomarker for neurodegeneration, was lower at six months following intrathecal MSC injection as compared to baseline levels, which also indicate a beneficial effect [132]. A long-term follow-up study showed that 22 of 24 patients were stable or improved at the last follow-up visit [133]. However, only seven patients completed the four-year endpoint, raising questions about the long-term findings’ validity.

Despite these promising results, more data from randomized trials are needed, especially in patients with progressive MS without active disease, for which there is no treatment available to prevent neural degeneration. However, the lack of knowledge concerning the fate and function of MSCs after intrathecal administration may prevent a successful clinical translation. Still, we cannot answer simple questions, like what happens to the cells once they are infused into the patient. How long do they survive and exert their function? If the MSCs are not integrated within the CNS tissue, which most pre-clinical studies suggest, the effect will likely be transient. Consequently, the transplantation must be repeated, decreasing its therapeutic value as the production of MSCs for clinical use is expensive and resource demanding. Recent trials have also shown that inflammatory reactions in the form of arachnoiditis can appear as a complication after intrathecal transplantation [134,135]. Because MSCs are highly secretory, the repeated administration of a cell-free secretome may be an alternative, as different studies have indicated in pre-clinical models. A cell-free product may also be injected in less invasive ways, such as intranasally. No clinical studies have so far explored these opportunities.

In conclusion, pre-clinical studies applying in vitro cultures and demyelinating rodent models have shown a great neuroprotective potential of MSCs. The most important mechanism appears to be related to their secretome, carrying cytokines, vesicles, and growth factors able to modulate pathogenic immune responses, promote remyelination, and slow axonal degeneration. Although clinical trials have shown promising results, primarily related to intrathecal administrations, there is not sufficient evidence to support the broader use of MSCs in MS. Their therapeutic use should be limited to clinical trials designed to assess efficacy compared to an active comparator or to answer questions related to their fate and therapeutic mechanisms in MS. Meanwhile, more pre-clinical research on priming the MSCs and “hand-tailoring” their secretome for the promotion of neuroregeneration seems to be an exciting avenue.
ijms-25-01365-t001_Table 1Table 1Published clinical trials assessing treatment with MSCs in multiple sclerosis.**First, Author, Country and Year****Condition and Important Inclusion Criteria****Timing of MSC Treatment after Debut of Condition****Design and Blinding****Follow-Up Time****Type of MSC & Administration****N Patients****N Controls****Main Results Safety****Main Results Efficacy****Controlled Studies**Li, China2014 [136]RRMS/SPMS EDSS 4–8 Treatment failure NR≥2 years+Randomized −Placebo −Blinded12 monthsAllogeneic MSCs from UC in combination with methylprednisolone IV × 31310No SARLower EDSS and relapse rate in MSC groupLlufriu, Spain2014 [56]RRMS EDSS 3–6.5 Treatment failure2–10 years+Randomized +Placebo +Double blinded +Cross-over6–12 monthsAutologous MSCs from BM IV × 199No SARTrend lower rate of CELLublin, USA2014 [137]RRMS/SPMS EDSS not specified Treatment failure≥2 years+Randomized +Placebo +Double blinded6–12 monthsAllogeneic, placenta-derived MSCs IV × 112 (6 low dose, 6 high dose)4One anaphylactoid reaction and one superficial thrombophlebitisNot assessed between groupsMeng, China2018 [138]MS type and EDSS not specified Treatment failureNS−Randomized −Placebo −Blinded1–3 yearsAllogeneic MSCs from UC IV × 721No SARNot assessed between groupsFernandez, Spain2018 [139]SPMS EDSS 5.5–9 Treatment failureNS+Randomized +Placebo +Double blinded12 monthsAutologous adipose-derived MSCs IV × 123 (11 low dose, 12 high dose)11No SARNo significant effectPetrou, Israel2020 [131]SPMS/PPMS EDSS 3–6.5 Treatment failure≥3 years+Randomized +Placebo +Double blinded +Cross-over6–12 monthsAutologous MSCs from BM IL and IV × 1–216 IT & 16 IV16No SARFewer patients with treatment failure and more patients with NEDA in MSC groupUccelli, Italy2021 [128]RRMS/SPMS/PPMS EDSS 2.5–6.5 Treatment failure NR2–15 years+Randomized +Placebo +Double blinded +Cross-over24–48 weeksAutologous MSCs from BM IV × 1144144No SARNo significant effect**Uncontrolled studies**Bonab, Iran2007 [140]Type MS NS EDSS ≤ 6 Treatment failureNS-12 monthsAutologous MSCs from BM IT × 1–210-Two iatrogenic meningitisOne EDSS improvement, stabile in four and worsening in fiveYamout, Lebanon2010 [7]MS type NS EDSS 4–7.5 Treatment failureNS-12 monthsAutologous MSCs from BM IT × 110-One transient encephalopathy with seizuresFive EDSS improvement, stabile in one and worsening in oneBonab, Iran2012 [8]SPMS/PPMS EDSS 3.5–7 Treatment failure2–15 years-12 monthsAutologous MSCs from BM IT × 125-No SAR reportedFour EDSS improvement, stabile in 12 and worsening in sixConnick, UK2012 [9]MS type not specified EDSS 2–6.5 Treatment failure NRNS-6 monthsAutologous MSCs from BM IV × 110-No SAR reportedImproved visual acuity and VEPOdinak, Russia2012 [10]MS type and EDSS NS Treatment failureNS-12 monthsAutologous MSCs from BM IV × 4–88-No SAR reportedSix EDSS improvement, stabile in one and worsening in oneHarris, USA2016 [11]SPMS/PPMS EDSS ≥ 3 Treatment failure NSNS-Mean 7.4 yearsAutologous MSCs from BM (differentiated in neural direction) IT × 2–56-No SAR reportedFour EDSS stable, worsening in twoDahbour, Jordan2017 [141]MS type & EDSS NS Treatment failureNS-12 monthsAutologous MSCs from BM IT × 210-No SAR reportedTwo EDSS improvement, stabile in four and worsening in fourCohen, USA2018 [142]RRMS/SPMS EDSS 3–6.5 Treatment failure NRNS-6 monthsAutologous MSCs from BM IV × 124-No SAR reported17 EDSS improvement, stabile in 8Harris, USA2018 [127]SPMS/PPMS EDSS ≥ 3 Treatment failure NSNS-12 monthsAutologous MSCs from BM (differentiated in neural direction) IT × 320-No SAR reportedEight EDSS improvement, stabile in ten and worsening in twoRiordan, Panama2018 [143]MS type NS EDSS 2–7 Treatment failure NRNS-12 monthsAllogeneic MSCs from umbilical cord IV × 720-No SAR reportedMean reduction of 0.68 in EDSS scoreSahraian, Iran2018 [144]RRMS/SPMS EDSS ≤ 5.5 Treatment failure2–15 years-2 yearsAutologous MSCs from BM IT × 1–26-
One EDSS improvement, stabile in two and worsening in threeIacobeus, Sweden2019 [145]RRMS/SPMS/PPMS EDSS 3–7 Treatment failure2–20 years-48 weeksAutologous MSCs from BM IV × 17-No SAR reportedNo significant changes in EDSS, one patient discontinued due to relapseCohen, USA2023 [135]PPMS EDSS 3–6.5 Treatment failure NRNS-28 weeksPre-modified MSCs from BM IT × 318-Two arachnoiditis (one patient discontinued)3 EDSS improvement, rates stable/worsening not reportedMSC: mesenchymal stem cells NS: not specified NR: not required EDSS: Expanded Disability Status Scale RRMS: relapsing-remitting multiple sclerosis SPMS: secondary progressive multiple sclerosis PPMS: primary progressive multiple sclerosis IV: intravenous IT: intrathecal.
ijms-25-01365-t002_Table 2Table 2Currently ongoing clinical trials with MSCs in multiple sclerosis.**Center (NCT-Number)****Important Inclusion Criteria****Timing of MSC Treatment****Design and Blinding****Primary Endpoint****Follow-Up Time****Type of MSC & Administration****N Patients****Estimated Study Completion**Bergen, Norway (NCT0474966)SPMS/PPMS EDSS4-7 Treatment failure NR2–15 years+Randomized +Placebo +Double blinded +Cross-overNeurophysiological parameters6 monthsAutologous BM-derived MSCs IT × 1182025St.John Antigua and Barbuda (NCT05003388)MSNS−Randomized −Placebo −BlindedSafety4 yearsAllogeneic MSCs from umbilical cord IV × 1152025Texas, United States (NCT04956744)RRMS EDSS 3–6.5 Treatment failure>6 months+Randomized +Placebo +Double blindedQuality of life1 yearAutologous adipose-derived MSCs IV × 6302023Atlanta, United States (NCT04956744)MS Treatment failureNS−Randomized −Placebo −BlindedSafety60 monthsAllogeneic embryonic MSCs IV × 1302027Taichung, Taiwan (NCT05532943)RRMS/SPMS EDSS 2–6.5 Treatment failure2–15 years−Randomized −Placebo −Blinded (second part with control group)First, part safety, second part efficacy1 yearAllogeneic umbilical cord MSCs IV and IT412026MSC: mesenchymal stem cells NS: not specified NR: not required EDSS: Expanded Disability Status Scale RRMS: relapsing-remitting multiple sclerosis SPMS: secondary progressive multiple sclerosis PPMS: primary progressive multiple sclerosis IV: intravenous IT: intrathecal.


## Figures and Tables

**Figure 1 ijms-25-01365-f001:**
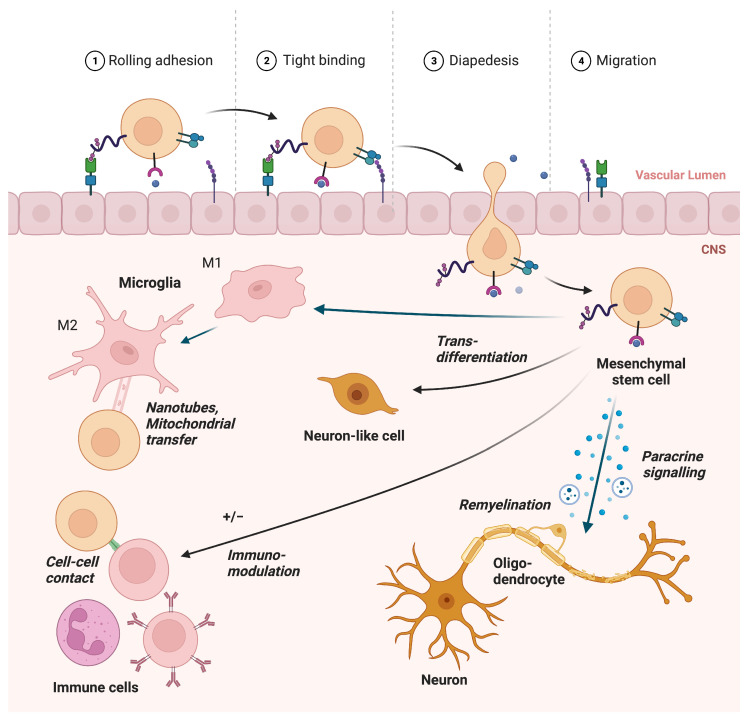
Potential regenerating mechanisms of MSCs in MS.

## Data Availability

Not applicable.

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
