# Peer review of "The Therapeutic Mechanisms of Mesenchymal Stem Cells in MS—A Review Focusing on Neuroprotective Properties"

_ijms, 2024, doi:10.3390/ijms25031365_

Round 1

Reviewer 1 Report

Comments and Suggestions for Authors

Pleasee see the attached file.

Comments on the Quality of English Language

To facilitate reader comprehension, it is recommended that the authors improve the quality of the manuscript's writing.

Author Response

The authors have provided a comprehensive overview of the current understanding of neuroprotective mechanisms. They also delved into the translation of mesenchymal stem cells (MSCs) transplantation and their derivatives from pre-clinical demyelinating models to clinical trials involving multiple sclerosis (MS) patients. The authors highlighted the substantial neuroprotective potential of MSCs in MS. A crucial mechanism appears to be linked to their secretome, which carries cytokines, vesicles, and growth factors capable of modulating pathogenic immune responses, promoting remyelination, and slowing axonal degeneration. I believe that readers in this field stand to benefit significantly from this work. However, some refinements are necessary before acceptance for publication. The following are a few suggested revisions:

– Thank you for very nice feedback. The suggestions are helpful and appreciated. We made the requested changes.

  1. Consider breaking down long sentences into shorter ones to enhance readability and comprehension.

    – We have rewritten parts of the text to improve readability.  

  2. Ensure a smooth flow between paragraphs and ideas, making it easier for readers to follow the progression of information.

    – We agree and have carefully edited the text and paragraphs for better flow.

  3. In the Introduction section, authors are suggested to rephrase Line 21 for clarity. Instead of "trigger for the pathogenic immunological events," you might say "plays a crucial role in the initiation of pathogenic immunological events."

    – This has been rephrased to:
    "[…] plays a crucial role in initiating pathogenic immunological events in MS."

  4. Since the manuscript mentions the importance of Epstein-Barr virus in triggering immunological events in MS, consider providing a brief explanation to studies supporting this claim for readers who may not be familiar with this aspect.

    – We have now expanded the text to support the claim:  
    “Recent data show that infection with the Epstein-Barr virus (EBV) plays a crucial role in initiating pathogenic immunological events in MS \cite{Bjornevik2022}. In a large cohort comprising 10 million young adults, longitudinal data revealed that the risk of MS increased 32-fold after infection with EBV, but not after infection with other viruses. Molecular mimicry has also been identified between the EBV transcription factor EBV nuclear antigen 1 (EBNA1) and the CNS protein glial cell adhesion molecule (GlialCAM) \cite{Lanz2022}."

  5. Highlight specific outcomes that support the regenerative role of MSCs in MS. The exploration on stem cells has been mentioned in “Nanomicro Lett, 2021 Dec, 2;14(1):4. doi: 10.1007/s40820-021-00747-8. The authors are suggested to discuss it.

    – We have now highlighted this in the discussion:
    “Although some studies have shown findings suggesting in vivo implantation and transdifferentiation of MSCs, an increasing amount of evidence points to secretory functions and cell-to-cell communicating abilities as the most important mechanisms responsible for their therapeutic impact. Through these mechanisms, MSCs contribute indirectly to endogenous neuroprotective and regenerative processes. Specifically, MSCs have been shown to promote CNS regeneration via modulation of microglia, remyelination by enhanced differentiation of oligodendrocytes and improved neural survival and outgrowth.”    

  6. Establish a clearer connection between in vitro findings, such as the effects of MSCs on peripheral blood mononuclear cells from MS patients, and clinical outcomes. More details may be found in recent work, such as: Smart Medicine, 2022 Dec, 1(1): e2022001427, doi:10.1002/SMMD.20220014. Help readers understand how in vitro observations align with or inform the clinical implications.

    – We agree that the connection could be clearer. We have rewritten this part of the article to make it clearer.

  7. In the Mechanisms section, when presenting contrasting results in the EAE and cuprizone models, briefly discuss the potential reasons for the discrepancies. This could include variations in study design, MSC sources, or administration methods.

    – We have now included these potential reasons:
    “Other studies have, however, shown negative results regarding remyelination in both EAE \cite{Glenn2015} and cuprizone \cite{Nessler2013} models after treatment with MSCs. Potential reasons for these discrepancies include differences in mode, type, dosage, and timing of MSC administration. The results may also highlight the heterogeneous nature of these stem cells.”  

  8. Emphasize the relevance of findings from the EAE model, where MSC-derived exosomes containing miRNA-367-3p alleviate disease progression by suppressing ferroptosis in microglia. Connect this observation to the characteristics of chronic active MS lesions and the cuprizone MS animal model.

    – We have now made the connections between ferroptosis and chronic active lesions more clear.

Reviewer 2 Report

Comments and Suggestions for Authors

As a reviewer, it is my pleasure to be able to evaluate manuscripts like this one, presented by Dr Sonia Gavasso et al, entitled: Therapeutic Mechanisms of Mesenchymal Stem Cells in MS—A Review Focusing on Neuroprotective Properties.

It is very well structured but I am missing a detail, the title talks about the therapeutic mechanisms of mesenchymal cells in multiple sclerosis, so it is necessary to develop the mechanisms involved, for example in the document I miss the following:

• A section on MSC exosomes and their cargo (a table)

• Greater detail in the secretome of MSCs in culture (since it contains exosomes and secret factors to the medium)

• Effect of MSCs on the polarization of inflammation, a little is discussed and more detail is needed due to the importance of inflammatory processes in MS.

Author Response

As a reviewer, it is my pleasure to be able to evaluate manuscripts like this one, presented by Dr Sonia Gavasso et al, entitled: Therapeutic Mechanisms of Mesenchymal Stem Cells in MS—A Review Focusing on Neuroprotective Properties.

It is very well structured but I am missing a detail, the title talks about the therapeutic mechanisms of mesenchymal cells in multiple sclerosis, so it is necessary to develop the mechanisms involved, for example in the document I miss the following:

Thank you for the constructive feedback on our article. We have implemented changes to several sections. 

  1. A section on MSC exosomes and their cargo (a table)
    and   
  2. Greater detail in the secretome of MSCs in culture (since it contains exosomes and secret factors to the medium)

    – We find it difficult to make a comprehensive table because the cargo is so variable and method-specific. We have implemented a section about MSC exosomes and rewritten parts of the secretome section highlighting cargo relevant for CNS repair:
    "The paracrine functions of the MSCs are mediated through secreted molecules, collectively named the \textit{secretome}, which have been shown to contain several diffusible biomolecules that promote the development, maintenance, repair, and survival of neuronal populations \cite{SamperAgrelo2020}. The soluble fraction contains cytokines and chemokines, such as IL-10, IL6 and CXCL-10 and growth factors, including GDNF, FGF, IGF and BDNF \cite{Pinho2020}. Beyond the soluble molecules, the secretome also contains extracellular vesicles with cargo such as proteins, nucleic acids, lipids and metabolites. The components of the secretome, and their effect on the environment, may vary depending on the source of MSCs and if pre-conditioning strategies have been applied.

    In an EAE mouse model, applying the secretome of stem cells from human exfoliated deciduous teeth (SHED) led to improved disability scores and a reduction of inflammation, demyelination, and axonal injury \cite{Shimojima2016}. The SHED secretome effectively inhibited T cell proliferation and reduced the production of pro-inflammatory cytokines. In addition, the infiltrating macrophages shifted from a pro-inflammatory phenotype to a pro-regenerative phenotype, thereby improving outcomes. A recent in vitro study also showed that the secretome from MSC-derived neural progenitor cells reduced the expression of pro-inflammatory markers in activated microglia \cite{Harris2023}. Other in vitro studies have revealed that the paracrine function of MSCs influences the destiny of neural stem cells by enhancing oligodendrogenesis and neurogenesis \cite{Bai2006, Bai2009}. This may lead to increased remyelination in MS. 

    A sub-group of the extracellular vesicles in the secreome is called \textit{exosomes}. These membrane-coated particles are 30-150 nm in diameter and transport different proteins and nucleic acids, serving as paracrine mediators. More than 300 proteins and 150 microRNAs have been identified in the exosomes of MSCs, in addition to other biomolecules \cite{Lotfy2023}. The exosomes may be isolated by using ultracentrifugation in combination with differential centrifugation or cross-flow filtration \cite{Jakl2023}. Exosomes may fully recapitulate, and even improve, the therapeutic effects of MSCs with regards to immunomodulation, stimulation of neurogenesis, and inhibition of apoptosis \cite{Giovannelli2023}. Exosomes have also been shown to have a modulatory effect on activated microglia in MS animal models. In a study using the EAE model, mice receiving intravenously administered exosomes had decreased activation of microglia and reduced inflammation and demyelination, resulting in improved functional outcomes \cite{Li2019_2}. In a similar study applying the EAE disease model, exosomes from MSCs were administrated intranasally. Results showed that nasal exosome treatment decreased CNS inflammation more effectively than treatment with MSCs, leading to better amelioration of the disease \cite{Fathollahi2021}."

  3. Effect of MSCs on the polarization of inflammation, a little is discussed and more detail is needed due to the importance of inflammatory processes in MS.

    More detail has been added to each of the sections related to immunology and inflammation.

Round 2

Reviewer 1 Report

Comments and Suggestions for Authors

accept.